# Effect of rotenone-induced stress on physiologically active substances in adult *Aphis glycines*

**Lanlan Han**[☯], **Litong Gao**[☯], **Ziru Hao, Kuijun Zhao**[ORCID]*, **Wenlin Zhang, Juan Chen, Jianfei Xiao, Aonan Zhang, Zhenghao Shi, Lin Zhu**

Agricultural Insect and Pest Control Task Group, College of Agriculture, Northeast Agricultural University, Harbin, Heilongjiang, China

☯ These authors contributed equally to this work.
* kjzhao@neau.edu.cn

**Data Availability Statement:** All relevant data are within the manuscript and its Supporting Information files.

## Abstract

The aim of this study was to determine the effect of rotenone stress on *Aphis glycines* Matsumura (Hemiptera: Aphididae) populations in different habitats of Northeast China. The changes in kinase expression activity of endogenous substances (proteins, total sugars, trehalose, cholesterol, and free amino acids), detoxifying enzymes (cytochrome P450 and glutathione S-transferase), and metabolic enzymes (proteases and phosphofructokinases) in specimens from three populations were compared before and after stress with rotenone at median lethal concentration ($LC_{50}$) and their response mechanisms were analyzed. Following a 24 h treatment with rotenone, the average $LC_{50}$ rotenone values in *A. glycines* specimens from field populations A and B, and a laboratory population were 4.39, 4.61, and 4.03 mg/L, respectively. The degree of changes in the kinase expression activity of endogenous substances also differed, which indicated a difference in the response of *A. glycines* specimens from varying habitats to $LC_{50}$ rotenone stress. The content of endogenous substances, detoxifying enzymes, and metabolic enzymes, except for that of free amino acids, changed significantly in all populations treated with rotenone at $LC_{50}$ compared with that in the control ($P < 0.05$). The decrease in protein and trehalose content, and the obstruction of cholesterol transportation owing to decreased feeding in stressed individuals were the causes of *A. glycines* death after rotenone treatment. *Aphis glycines* resistance to rotenone may be related to cytochrome P450 expression.

## Introduction

*Aphis glycines* Matsumura (Hemiptera: Aphididae) is one of the main pests of soybeans and harms soybean plants by feeding on the leaves and causing undesirable effects, such as soybean leaf curling and plant dwarfing [1], which in turn leads to a series of economic problems such as decreased soybean yield and reduced quality [2]. Currently, the prevention and control of *Aphis glycines* is primarily based on chemical methods, but the abuse of chemical pesticides

**Funding:** This work was supported by Special Fund for Construction of Modern Agricultural Industry Technology System (grant number CARS-04); Heilongjiang Science Foundation Project (grant number C2018011).

**Competing interests:** The authors have declared that no competing interests exist.

has not only caused certain damage to the environment but has also resulted in pesticide-resistant *A. glycines* [3]. In 2004, in Mudanjiang (Heilongjiang Province, China), although the dosage of dimethoate used to control *A. glycines* has almost doubled, its efficacy is still declining [4]. Therefore, utilizing integrated pest management systems, making scientific and rational use of chemical pesticides, reducing damage to farmland ecosystems, and controlling harmful organisms below the allowable level of economic damage are the focus of recent studies [5].

Rotenone, extracted from the roots of leguminous plants, is a broad-spectrum plant insecticide. It causes stomach and contact toxicity, and acts as an antifeedant and a fumigant with control effects on the pests of 137 families in 15 orders [6, 7]. Rotenone can inhibit cell respiration; it is an electron transfer inhibitor that blocks electron transfer from nicotinamide adenine dinucleotide to coenzyme Q. Rotenone is a natural compound that can degrade quickly with low toxicity. It is a pesticide that can meet the needs of an ecologically aware civilization [6].

The processes of growth, development, metamorphosis, and reproduction of insects are inseparable from the synthesis, decomposition, and transformation of proteins, lipids, carbohydrates, and other substances in the insect body. The content of metabolic substances affects the growth and development of insects to a certain extent [8], thus reflecting the insect's ability to adapt to the environment. By measuring the changes in the content of metabolic substances in insects under the influence of external factors, it is possible to explore the internal relationships between various substances in insects.

In this study, we compared and analyzed the variable differences and trends in the protein, total sugar, trehalose, cholesterol, and free amino acid (FAA) content as well as protease, glutathione-S-transferase (GST), cytochrome P450 (CYP450), and phosphofructokinase (PFK) activities in adult *A. glycines* populations from three habitats under median lethal concentration ($LC_{50}$) rotenone stress to find a more efficient method to comprehensively control *A. glycines* and provide a theoretical basis for the effects of rotenone on this insect species.

## Materials and methods

### Insect sources

*Aphis glycines* adults were collected in the Xiangyang farm, Northeast Agricultural University Harbin, Heilongjiang, China. *Aphis glycines* individuals collected in a corn and soybean neighbor-cropping field were used as field population A and those collected in a potato and soybean neighbor-cropping field were used as field population B. During the peak season of *A. glycines* infestation in summer, 80–100 soybean leaves were collected from the two soybean fields, and 1–10 *A. glycines* specimens were collected from the back of each soybean leaf; this process was repeated five times. The *A. glycines* adults for the laboratory population were collected from soybean plants cultivated in an artificial climate chamber (ambient temperature: 24 ˚C; photoperiod 16L:8D; relative humidity: 60% ± 5%) in the laboratory, and had been cultured continuously for more than 3 years, the laboratory *A. glycines* population has been kept in the laboratory, according to the experimental needs, take the corresponding number of *A. glycines*.

### Determination of 24 h $LC_{50}$ rotenone treatment on *A. glycines*

We spread 1% agar medium in a 6 cm diameter plastic Petri dish and allowed it to solidify. We prepared 10 mL rotenone microemulsion) formulations at concentrations of 16, 8, 4, 2, 1, and 0.5 mg/L in 25 mL beakers. Fresh soybean leaves (approximately 1.5 cm$^2$ per piece) were immersed in each of the five prepared solutions for 2 s, taken out, then pasted onto the prepared medium, and labeled; tape water was used for immersing the control group. One prepared leaf was placed on the agar medium in each Petri Dish, and the *A. glycines* specimens

were inserted into the Petri dish after the leaf had dried; four specimens from the same population were placed onto each soybean leaf. Each concentration required 20 specimens and the experiment was repeated three times for each concentration, thus 360 individuals from each population were required for the experiments. After 24 h, the specimens were observed under a dissecting microscope as follows: the specimens were gently touched with a writing brush and observed for movement. If the specimen moved within 3–5 s, it was recorded as alive. This procedure was repeated with each insect. The effect of the 24 h $LC_{50}$ rotenone treatment on the *A. glycines* specimens was calculated from the death rate obtained at the end of each 24 h cycle.

### Determination of metabolic substance content in *A. glycines*

*Aphis glycines* adults were stressed with rotenone $LC_{50}$ for 24 h using the same leaf dipping method as explained previously. After the stress treatment, live *A. glycines* specimens of similar size were collected from the treatment and control groups, and 10 insects were placed in a 1.5 mL Eppendorf (EP) tube, cooled in liquid nitrogen, and refrigerated at -80 ˚C for future use. Seventy *A. glycines* adults of similar size were treated in each experiment. Thirty live insects were collected 24 h later, and the experiment was repeated several times until all physiological indicators were measured. These collected *A. glycines* were placed on ice blocks, and after adding 300 μL of phosphate buffer saline (PBS) buffer, they were uniformly ground and mixed. The tissue homogenate was then centrifuged at $8000 \times g$ for 10 min. The supernatant was transferred into a new EP tube and the content of physiologically active substances in *A. glycines* was determined using biochemical kits. The protein, total sugar, trehalose, and cholesterol content, and the PFK and GST activity were detected by using commercial assay kits obtained from Beijing Solarbio Science & Technology Co., Ltd. (Beijing, China). The Insect Free Amino Acid (FAA) ELISA Kit, Insect cytochrome P450 (CYP450) ELISA kit, and Insect protease (Pro) ELISA kit were obtained from Jiangsu Meibiao Biotechnology Co., Ltd. (Yancheng, China).

### Data analysis

Statistical analysis software SPSS23.0 was used for data analyses. Independent sample *t*-test was adopted to compare the significant difference in physiologically active substances before and after chemical treatment. By using analysis of variance (ANOVA) combined with the least significant difference (LSD) method, multiple comparisons were made to analyze significant differences in physiologically active substances among the three populations, and the level of significance was $P < 0.05$.

## Results

### Determination of the toxicity of rotenone against *A. glycines*

As shown in Table 1, the highest average $LC_{50}$ value of rotenone in *A. glycines* tissues was obtained from field population B at 4.61 mg/L, followed by field population A at 4.39 mg/L, and the least was from the laboratory population at 4.03 mg/L.

### Influence of rotenone $LC_{50}$ stress on the content of protein metabolism-related substances in *A. glycines* adults of three populations

Compared with that in the control, after being stressed with rotenone for 24 h at $LC_{50}$, the content of protein decreased significantly in field population A (F = 4.088, df = 4, P = 0.003), field population B (F = 10.469, df = 2.107, P = 0.024), and in the laboratory population (F = 1.439,

**Table 1. Toxicity test results.**

| Testing population | LC$_{50}$ value (mg/L) | Toxicity regression equation | Correlation coefficient |
| --- | --- | --- | --- |
| Field population A | 4.39 (3.9423–4.8874) | Y = 3.9151 + 1.6888x | 0.9963 |
| Field population B | 4.61 (3.2897–6.4568) | Y = 3.8905 + 1.6720x | 0.9657 |
| Laboratory population | 4.03 (2.5626–6.3393) | Y = 4.0696 + 1.5369x | 0.9366 |

df = 4, P = 0.017) by 28.4%, 15.0%, and 20.3%, respectively. The activity of protease increased significantly in field population A (F = 6.031, df = 4, P = 0.033), field population B (F = 0.437, df = 4, P = 0.009), and in the laboratory population (F = 2.578, df = 4, P < 0.0001) by 26.5%, 41.3%, and 92.1%, respectively, whereas the FAA content increased in field population A (F = 3.906, df = 4, P = 0.072), field population B (F = 0.331, df = 4, P = 0.268), and in the laboratory population (F = 7.180, df = 4, P = 0.189) by 16.5%, 9.48%, and 19.8%, respectively (Fig 1), but was not significant.

## Influence of rotenone LC$_{50}$ stress on the content of sugar metabolism-related substances in *A. glycines* adults of three populations

Compared with that in the control, after being stressed with rotenone for 24 h at LC$_{50}$, the content of total sugar decreased by 2.71%, 4.47%, and 1.88% in field population A (F = 10.302, df = 2.134, P = 0.125), field population B (F = 0.047, df = 4, P = 0.06), and in the laboratory population (F = 1.225, df = 4, P = 0.193), respectively, but was only significant in field population B. The trehalose content decreased significantly by 21.3%, 14.5%, and 38.4% in field population A (F = 0.072, df = 4, P = 0.033), field population B (F = 1.323, df = 4, P = 0.015), and in the laboratory population (F = 3.484, df = 4, P = 0.048), respectively, whereas the PFK activity increased significantly by 99.1%, 68.5%, and 53.2% in field population A (F = 2.691, df = 4,

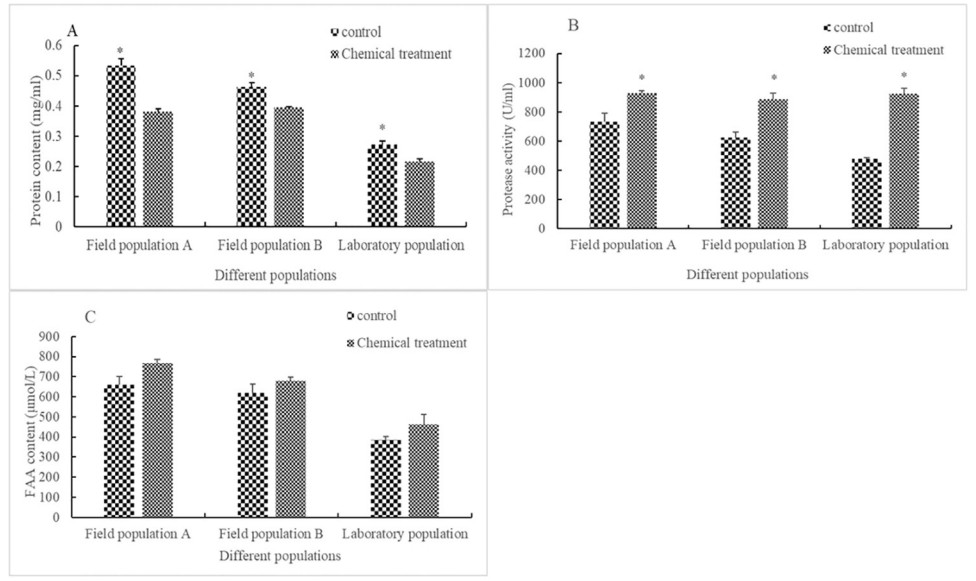

**Fig 1. Influence of LC$_{50}$ rotenone on protein metabolism in different populations of *Aphis glycines*.** The influence of LC$_{50}$ rotenone on protein content (A), protease content (B), and FAA content (C) in different populations of *Aphis glycines*. Different stripes in bars represent different treatments. * indicates that there was a significant difference (*P* < 0.05) between the chemical treatment group and control check group for the adults of *A. glycines* of the same population. Bar heights represent the sample mean and error bars are the standard error of the means.

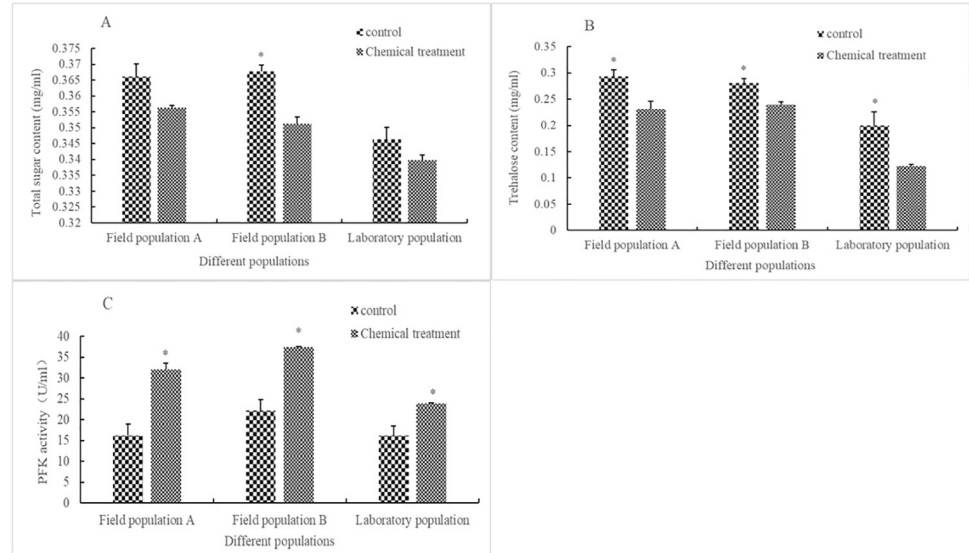

**Fig 2. Influence of LC$_{50}$ rotenone on sugar metabolism in different populations of *Aphis glycines*.** The influence of LC$_{50}$ rotenone on total sugar content (A), trehalose content (B), and PFK activity (C) in different populations of *A. glycines*. Different stripes in bars represent different treatments. * indicates that there was a significant difference ($P < 0.05$) between the chemical treatment group and control check group for the adults of *A. glycines* of the same population. Bar heights represent the sample mean and error bars are the standard error of the means.

$P = 0.006$), field population B ($F = 9.414$, $df = 2.020$, $P = 0.027$), and in the laboratory population ($F = 7.620$, $df = 4$, $P = 0.032$), respectively (Fig 2).

### Influence of rotenone LC$_{50}$ stress on the content of several metabolic substances in *A. glycines* adults of three populations

Compared with that in the control, after being stressed with rotenone for 24 h at LC$_{50}$, the content of cholesterol increased significantly in field population A ($F = 7.651$, $df = 4$, $P = 0.036$), field population B ($F = 2.936$, $df = 4$, $P = 0.013$), and in the laboratory population ($F = 1.517$, $df = 4$, $P = 0.037$) by 39.2%, 69.7%, and 32.7%, respectively. The activity of GST decreased significantly in field population A ($F = 2.110$, $df = 4$, $P = 0.032$), field population B ($F = 0.036$, $df = 4$, $P = 0.007$), and in the laboratory population ($F = 12.430$, $df = 4$, $P = 0.003$) by 22.6%, 18.9%, and 71.6%, respectively. The content of CYP450 increased by 14.8%, 24.2%, and 76.7% in field population A ($F = 1.123$, $df = 4$, $P = 0.135$), field population B ($F = 0.693$, $df = 4$, $P = 0.051$), and in the laboratory population ($F = 0.258$, $df = 4$, $P = 0.006$), respectively (Fig 3), but only increased significantly in the laboratory population.

### Comparison of the content of main metabolic substances in *A. glycines* adults of three populations before rotenone LC$_{50}$ stress

As shown in Table 2, the content of protein in *A. glycines* adults before the stress was the highest in field population A at 0.53 mg/mL and lowest in the laboratory population at 0.27 mg/mL; the difference among the three populations was significant ($P < 0.05$). The activity of protease in population A was the highest at 733.89 U/mL, and that in the laboratory population was the lowest, at 481.83 U/mL, with a significant difference ($P < 0.05$) between either field population (A or B) and laboratory population. The content of FAA was the highest in field

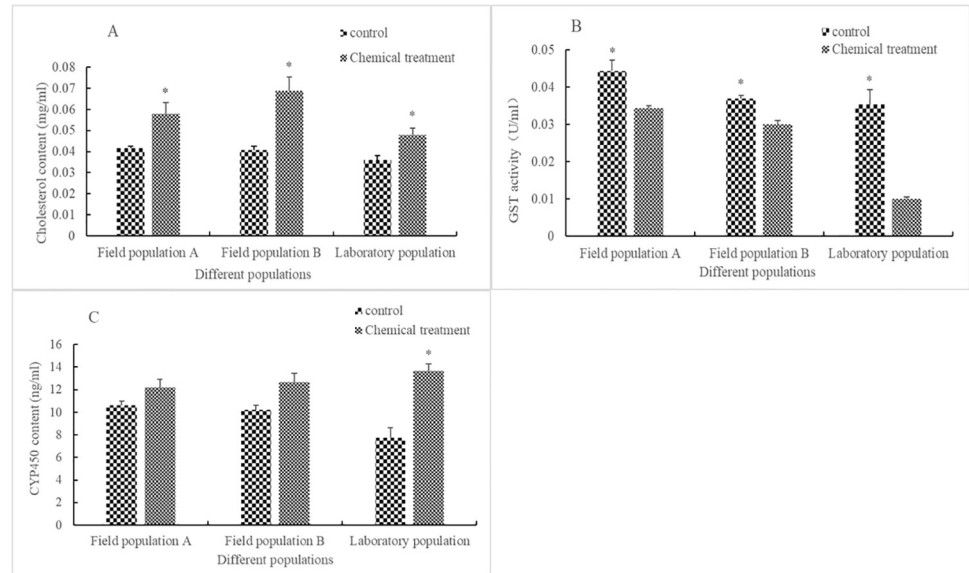

**Fig 3. Influence of LC$_{50}$ rotenone on other metabolic substances in different populations of *Aphis glycines*.** The influence of LC$_{50}$ rotenone on the cholesterol content (A), GST activity (B), and CYP450 content (C) in different populations of *A. glycines*. Different stripes in bars represent different treatments. * indicates that there was a significant difference ($P < 0.05$) between the chemical treatment group and control check group for *A. glycines* adults of the same population. Bar heights represent the sample mean and error bars are the standard error of the means.

population A at 659.30 μmol/L and lowest in the laboratory population at 384.82 μmol/L, with a significant difference ($P < 0.05$) between either field population (A or B) and laboratory population.

The content of total sugar in *A. glycines* was the highest in field population B at 0.368 mg/mL and lowest in the laboratory population at 0.346 mg/mL, with a significant difference ($P < 0.05$) between either field population (A or B) and laboratory population. The content of trehalose was the highest in population A at 0.29 mg/mL and lowest in the laboratory population at 0.20 mg/mL, with a significant difference ($P < 0.05$) between either field population (A or B) and laboratory population. The activity of PFK was the highest in field population B at 22.22 U/mL and lowest in the laboratory population at 16.12 U/mL, without significant difference among the three populations.

The content of cholesterol in *A. glycines* was the highest in field population A at 0.042 mg/mL and lowest in the laboratory population at 0.036 mg/mL, without significant difference among the three populations. The activity of GST was the highest in field population A at 0.044 U/mL and lowest in the laboratory population at 0.035 U/mL, with no significant difference among the three populations. The content of CYP450 was the highest in field population A at 10.61 ng/mL and lowest in the laboratory population at 7.72 ng/mL, with a significant difference ($P < 0.05$) between either field population (A or B) and laboratory populations.

## Discussion

The toxicity test results showed that *A. glycines* individuals from the laboratory population were the most sensitive to rotenone, followed by those from field population A and field population B. The LC$_{50}$ value of individuals from field populations A and B was 1.09- and 1.14-fold higher that of the laboratory population, respectively, indicating that *A. glycines* adults of these field populations had not developed resistance to rotenone. The results of the rotenone stress

**Table 2. Content/activity of physiological indexes in *Aphis glycines* of field population A, field population B, and laboratory population before treatment with LC$_{50}$ rotenone.**

| Physiologically active substance | Insect source | Content/activity |
|---|---|---|
| Protein (mg/mL) | Field population A | 0.53 ± 0.022a |
| | Field population B | 0.46 ± 0.012b |
| | Laboratory population | 0.27 ± 0.012c |
| Protease (U/mL) | Field population A | 733.89 ± 58.680a |
| | Field population B | 626.59 ± 34.104a |
| | Laboratory population | 481.83 ± 6.791b |
| FAA (μmol/L) | Field population A | 659.30 ± 41.329a |
| | Field population B | 620.47 ± 41.352a |
| | Laboratory population | 384.82 ± 17.764b |
| Total sugar (mg/mL) | Field population A | 0.366 ± 0.004a |
| | Field population B | 0.368 ± 0.002a |
| | Laboratory population | 0.346 ± 0.004b |
| Trehalose (mg/mL) | Field population A | 0.29 ± 0.012a |
| | Field population B | 0.28 ± 0.009a |
| | Laboratory population | 0.20 ± 0.027b |
| PFK (U/mL) | Field population A | 16.12 ± 2.779a |
| | Field population B | 22.22 ± 2.587a |
| | Laboratory population | 16.21 ± 2.328a |
| Cholesterol (mg/mL) | Field population A | 0.042 ± 0.001a |
| | Field population B | 0.041 ± 0.002a |
| | Laboratory population | 0.036 ± 0.002a |
| GST (U/mL) | Field population A | 0.044 ± 0.003a |
| | Field population B | 0.037 ± 0.001a |
| | Laboratory population | 0.035 ± 0.004a |
| CYP450 (ng/mL) | Field population A | 10.61 ± 0.376a |
| | Field population B | 10.18 ± 0.441a |
| | Laboratory population | 7.72 ± 0.908b |

In this table, the content/activity of a substance followed by a different letter indicates a significant difference ($P < 0.05$) between these populations; the data in the table are the mean ± standard error.

test showed that the variable quantity of physiologically active substances in the laboratory population changed the most, indicating that the resistance of soybean aphids in laboratory population to the stress of rotenone was the weakest, followed by that of individuals from field population A and field population B, that is, different populations in different cropping patterns had a difference in resistance to rotenone. The analysis of metabolic substances in *A. glycines* adults before being stressed with LC$_{50}$ rotenone in the three populations revealed higher content in the field population than in the laboratory population, indicating that *A. glycines* in field had developed a corresponding adaptability after long-term interspecific competition along with the climatic stress and other factors. The results of this study showed that *A. glycines* in field with different cropping patterns had differing susceptibility to pesticides. This result can provide guidance for the use of precise doses in fields, thereby reducing cost and pollution at the same time (Fig 4).

The results showed that after a 24 h stress with LC$_{50}$ rotenone, the activity of protease increased significantly ($P < 0.05$) in *A. glycines* adults in all the three populations, the content

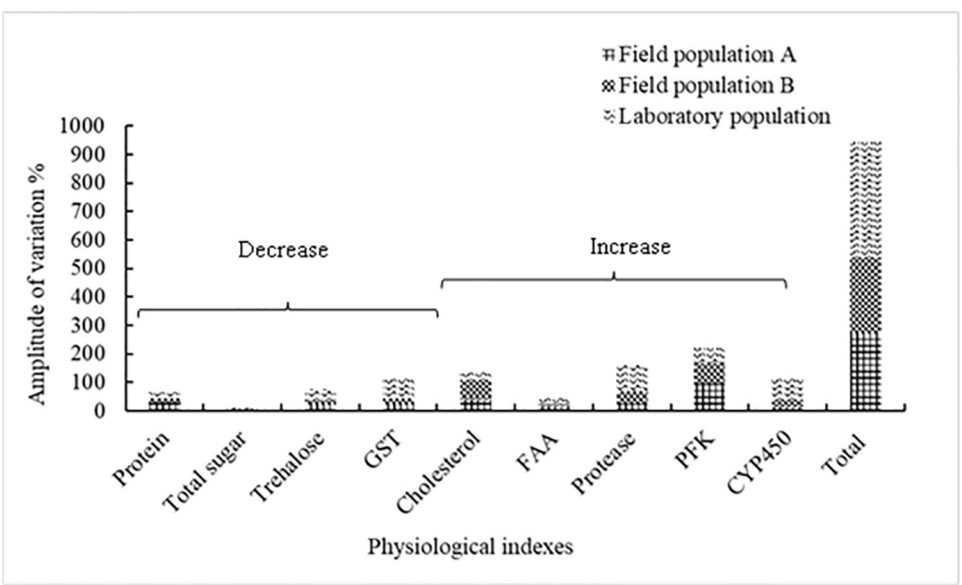

**Fig 4. Stacking diagram of physiological indexes of different populations of *Aphis glycines*.** Different stripes in bars represent different populations. The first four bars indicate the decrease in content/activity, the next five bars indicate the increase in content/activity, and the last bar indicates the total change range.

of protein decreased significantly ($P < 0.05$), and the content of FAA showed no significant difference. This indicated that protein decomposition was accelerating, most amino acids produced after decomposition were involved in the synthesis of new proteins to maintain normal living activities, and a small part of the amino acids continued to be free in the hemolymph to maintain the balance of blood osmotic pressure of the insect [9, 10, 11, 12, 13]. The trehalose content decreased significantly ($P < 0.05$) in the three populations and the activity of PFK increased significantly ($P < 0.05$); the total sugar content decreased in all the three populations but the decrease was significant ($P < 0.05$) only in field population B, implying the conversion of other substances into sugar, while the glycolysis rate was enhanced to maintain the stability of total sugar content in *A. glycines* [14, 15, 16]. Protein metabolism and sugar metabolism played an important role in the resistance of *Aphis glycines* to the 24 h stress of rotenone. This result is consistent with Cheng Weixia's study results on the preference of chosen polysaccharides and soluble proteins as metabolites when *Liposcelis entomophila* Enderlein (Psocoptera, Liposcelididae, *Liposcelis*) and *Liposcelis bostrychophila* Badonnel (Psocoptera, Liposcelididae, *Liposcelis*) were resistant to poor environments [17].

The results showed that after a 24 h stress with rotenone at $LC_{50}$, the cholesterol content increased significantly ($P < 0.05$) in *A. glycines* adults in the three populations. Simultaneously, *A. glycines* adults' feeding capability was weakened and the possibility of receiving more cholesterol from food was low. Therefore, the most likely reason is that cholesterol transportation in *A. glycines* was blocked, which ultimately affected the growth and development of *A. glycines* [18, 19, 20, 21]. The activity of GST was weakened significantly ($P < 0.05$) in the three populations, whereas the content of CYP450 increased but the increase was significant ($P < 0.05$) only in the laboratory population, indicating that CYP450 in *A. glycines* played a role in the response to rotenone stress, presumably due to the increased expression of this gene and may be related to the development of resistance [22, 23]. Previous studies have shown that the resistance of mosquitoes to insecticides was related to the increase in the expression of CYP450

[24], suggesting that we should pay attention to the changes in this gene when using rotenone in the future. The weakened activity of GST may be caused by increased consumption or by the inhibitory effect of rotenone, but the specific reason for this phenomenon needs further study [25, 26, 27].

In summary, after being stressed with rotenone at $LC_{50}$, *A. glycines* showed some effects on some physiological factors, such as the decrease in protein and trehalose content, and blocking of cholesterol transport, that resulted in their metabolic imbalance, slowed movement, and eventual death. *Aphis glycines* of different populations from different cropping patterns showed a difference in resistance and adaptability to rotenone, therefore, their controlling methods should be adjusted according to the cropping patterns to achieve the goal of effective pest control and reduced environmental pollution.

## Supporting information

**S1 Dataset.**
(XLSX)

## Author Contributions

**Data curation:** Litong Gao.

**Methodology:** Lanlan Han, Ziru Hao, Wenlin Zhang, Juan Chen, Jianfei Xiao, Aonan Zhang, Zhenghao Shi, Lin Zhu.

**Project administration:** Kuijun Zhao.

**Writing – original draft:** Lanlan Han, Litong Gao.

**Writing – review & editing:** Kuijun Zhao.

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
