## [Decision Letter · Decision Letter 0]

30 Mar 2020

PONE-D-19-35139

Effect of rotenone-induced stress on physiologically active substances in adult Aphis glycines

PLOS ONE

Dear Prof. Zhao,

Thank you for submitting your manuscript to PLOS ONE. After careful consideration, we feel that it has merit but does not fully meet PLOS ONE’s publication criteria as it currently stands. Therefore, we invite you to submit a revised version of the manuscript that addresses the points raised during the review process.

We would appreciate receiving your revised manuscript by May 14 2020 11:59PM. Please respond to all the reviewer(s) comments when revising the manuscript. (Note that reviewer's comments are embedded in the attached .pdf copy of the manuscript).

To enhance the reproducibility of your results, we recommend that if applicable you deposit your laboratory protocols in protocols.io, where a protocol can be assigned its own identifier (DOI) such that it can be cited independently in the future. For instructions see: http://journals.plos.org/plosone/s/submission-guidelines#loc-laboratory-protocols

We look forward to receiving your revised manuscript.

Kind regards,

Jed N. Lampe, Ph.D.

Academic Editor

PLOS ONE

Journal Requirements:

Reviewers' comments:

Reviewer's Responses to Questions

**Comments to the Author**

1. Is the manuscript technically sound, and do the data support the conclusions?

Reviewer #1: Yes

2. Has the statistical analysis been performed appropriately and rigorously? 

Reviewer #1: Yes

3. Have the authors made all data underlying the findings in their manuscript fully available?

Reviewer #1: Yes

4. Is the manuscript presented in an intelligible fashion and written in standard English?

Reviewer #1: No

5. Review Comments to the Author

Reviewer #1: The manuscript, Effect of rotenone-induced stress on physiologically active substances in adult Aphis glycines is a good work. The authors study the effect of rotenone insecticides on soyabean aphid. The English of manuscript is very poor, should be improved by native speaker. In Material and method section, some parts need clarity. According to my view, the manuscript can be published with major revision. My comments are on PDF file attached.

6. PLOS authors have the option to publish the peer review history of their article (what does this mean?). If published, this will include your full peer review and any attached files.

Reviewer #1: No

---

## [Author Response · Author response to Decision Letter 0]

17 May 2020

RESPONSES TO THE ACADEMIC EDITOR:

Comment 1: I have made this change to maintain consistency. (new line 23)

Answer: We thank you for your suggestion and accept that the change helps to maintain consistency.

Comment 2: All abbreviations should be defined at the first mention in the main text. (new line 29)

Answer: Thank you very much for pointing out this discrepancy. We have also noticed this oversight on our part and have made the relevant corrections. We believe this will also be helpful for our future writing.

Comment 3: A unit need not be repeated with all the numbers in a series or range. (new line 31)

Answer: Thank you for this tip; we will pay attention to this issue in our future writing.

Comment 4: After the first mention, the genus name should be abbreviated. (new line 33)

Answer: Thank you very much for your explanation and we apologize for this oversight on our part. We have made the appropriate changes in the manuscript, except at the beginning of sentences where the names have been spelled out.

Comment 5: Provide a space before and after mathematical signs. (new line 36)

Answer: Thank you very much for your tip. We apologize for not paying attention to this issue and believe that it will not appear in our future writing.

Comment 6: Please mention the city and country of all the manufacturing companies mentioned in the Materials and Methods section. If the manufacturer is US-based, the city and state generally suffice. For a vendor outside the USA: Description (Product name; Company name, City/Town, Country) Or, Product name (Company name, City/Town, Country). Please maintain consistency when providing the vendor information. (new line 118)

Answer: We thank you for your comment. We have marked the Product name and origin (Company name, City/Town, Country) clearly in the text according to your suggestion.

“The protein, total sugar, trehalose, and cholesterol content, and the PFK and GST activity was detected by using commercial assay kits obtained from Beijing Solarbio Science & Technology Co., Ltd. (Beijing, China). The Insect Free Amino Acid (FAA) ELISA Kit, Insect cytochrome P450 (CYP450) ELISA kit, and Insect protease (Pro) ELISA kit were obtained from Jiangsu Meibiao Biotechnology Co., Ltd. (Yancheng, China).”

Comment 7: Avoid beginning headings and titles with an article. (new line 133)

Answer: We thank you for your astute observation, and have revised the title of the table to “Toxicity test results.”

Comment 8: Although the conjunctions 'while' and 'whereas' have similar uses, there are some differences too. 'While', for example, can be used to events occurring simultaneously, but 'whereas' cannot be. The latter is generally preferred when highlighting differences between two things. (new line 144)

Answer: We thank you very much for your patience in explaining these minute differences to us. We now have a better understanding of the words “while” and “whereas” and will endeavor to use them accordingly.

Comment 9: Ordinal adjectives such as “first,” “fifth,” and “last” are typically preceded by the definite article “the.” (new line 197)

Answer: Thank you very much for your patience in explaining the usage of the article "the". We will pay attention to this problem and avoid making such mistakes in the future.

Comment 10: Between vs. among. Use "between" for two entities, "among" for more than two. (new line 199)

Answer: Thank you very much for your patience in explaining the difference between the words “between” and “among”. We will pay attention to this detail and try to avoid repeating these mistakes in the future.

Comment 11: What does “its” refer to? (new line 268)

Answer: Here, "its" refers to cholesterol, and we have modified the sentence for clarity.

RESPONSES TO THE REVIEWER:

Comment 1: Provide taxonomic detail (new line 2)

Answer: We thank you for your comments and appreciate the time taken by you for reviewing our manuscript. The change has been revised in the text according to your prompt. However, considering the format/style of the journal, we may have to make final modifications based on the joint opinions of both parties.

Comment 2: Provide complete information, authority, order, family (new line 24)

Answer: Thank you very much for your comments, we have revised it to “Aphis glycines Matsumura (Hemiptera: Aphididae).”

Comment 3: detoxifying enzymes (cytochrome P450 , glutathione-S-transferase, proteases and phosphofructokinases) (new line 26)

Answer: We thank you very much for your suggestion and your detailed comment. However, after discussion, we believe that, although protease and PFK may also be involved in detoxification, it is not appropriate to categorize these two enzymes as detoxifying enzymes but, instead, as metabolic enzymes.

Comment 4: Rephrase the sentence, unable to understand (new line 30)

Answer: We apologize for the unclear language and have revised it to – 

“Following a 24 h treatment with rotenone, the average LC50 rotenone values in A. glycines specimens from field populations A and B, and the laboratory population were 4.39, 4.61, and 4.03 mg/L, respectively. ”

Comment 5: This unit is not possible, it may be mg/L (new line 31)

Answer: We apologize for making such a mistake. After careful inspection, it was found that the unit was not clarified during the conversion process. We understand that this is a serious mistake, and we will pay great attention to such matters in future writing.

Comment 6: Authority? (new line 45)

Answer: After careful inspection, we believe that it is authoritative. However, in deference to your suggestion, we have modified it to 

“Aphis glycines Matsumura (Hemiptera: Aphididae).”

We hope this will be acceptable.

Comment 7: This paragraph is totally irrelevant to study, the authors provided here biological control of aphids, while the study is about chemical control. I suggest concise in one or two line and merge in first paragraph or delete it. (new line 56)

Answer: We have carefully considered your opinion and believe that your suggestion is reasonable. We have revised this paragraph but considering the context and the structure of the entire article, we believe that deleting this paragraph would be more conducive to the coherence of the article. We thank you for this keen observation.

Comment 8: how many adults were collected? (new line 80-88)

Answer: We explain this in detail in the following text, 

“During the peak season of A. glycines infestation in summer, 80–100 soybean leaves were collected from the two soybean fields, and 1–10 A. glycines specimens were collected from the back of each soybean leaf; this process was repeated five times.”

Comment 9: It means artificial climate chamber produced insects, how? is it possible growth chamber produce insects, i am really wonderful. Provide area from where susceptible population was collected. (new line 83-88)

Answer: We apologize for our previous statement which was unclear, and may have resulted in this misunderstanding. We have modified the sentence as follows for more clarity – 

“The A. glycines adults for the laboratory population were collected from soybean plants cultivated in an artificial climate chamber (ambient temperature: 24 ℃; photoperiod 16L:8D; relative humidity: 60% ± 5%) in the laboratory, and had been cultured continuously for more than 3 years.”

Comment 10: this is not effect of lc50, I think this procedure is to determine the LC50, be clear. (new line 89)

Answer: We thank you for your suggestion. We have modified this title to 

“Determination of 24 h LC50 rotenone treatment on A. glycines”.

Comment 11: clear water→tape water (new line 94)

Answer: We thank you very much for your suggestion. We have revised it according to your suggestion to “tap water.”

Comment 12: Clarity is missing, please rephrase the sentence. (new line 90-99)

Answer: We apologize for the unclear language in these sentences. We have revised it to 

“Fresh soybean leaves (approximately 1.5 cm2 per piece) were immersed in each of the five prepared solutions for 2 s, taken out, then pasted onto the prepared medium, and labeled; tap water was used for immersing the control group. One prepared leaf was placed on the agar medium in each Petri dish, and the A. glycines specimens were inserted into the Petri dish after the leaf had dried; four specimens from the same population were placed onto each soybean leaf. Each concentration required 20 specimens, and the experiment was repeated three times for each concentration, thus 360 specimens from each population were required for the experiment.”

Comment 13: Provide probit line of LC50 (new line 133)

Answer: In the list we give the virulence regression equation, it is the probit line of LC50.

Comment 14: Provide statistic in front of each population instead of together (new line 138-185)

Answer: We thank you for your valuable suggestion, and have revised the full text according to your direction. 

Comment 15: Kept two digits after decimal (new line 215)

Answer: We appreciate your attention to detail and thank you very much for your suggestion. We have made the modifications per your suggestion. However, for substances such as total sugar, there existed only a little difference among the three populations i.e. in the third decimal place, therefore we have retained three digits after the decimal point for this parameter.

Comment 16: This word is not correct for insecticide, replace with proper word. (new line 227)

Answer: Per your suggestion, we carefully reviewed the use of this word and believe that the use of “toxicity” is correct in this context, and have therefore modified the text.

Comment 17: Provide authority name (new line 262)

Answer: Thank you for your suggestion. Per your advice, we have revised the sentence to – 

“Liposcelis entomophila Enderlein (Psocoptera, Liposcelididae) and Liposcelis bostrychophila Badonnel (Psocoptera, Liposcelididae).”

---

## [Editor Report · Decision Letter 1]

20 May 2020

Effect of rotenone-induced stress on physiologically active substances in adult Aphis glycines

PONE-D-19-35139R1

Dear Dr. Zhao,

We are pleased to inform you that your manuscript has been judged scientifically suitable for publication and will be formally accepted for publication once it complies with all outstanding technical requirements.

With kind regards,

Jed N. Lampe, Ph.D.

Academic Editor

PLOS ONE
---

## [Editor Report · Acceptance letter]

26 May 2020

PONE-D-19-35139R1 

Effect of rotenone-induced stress on physiologically active substances in adult Aphis glycines 

Dear Dr. Zhao:

I am pleased to inform you that your manuscript has been deemed suitable for publication in PLOS ONE. Congratulations! Your manuscript is now with our production department. 

With kind regards,

on behalf of

Dr. Jed N. Lampe 

Academic Editor

PLOS ONE